# COVID-19 in Lung Transplant Recipients: A Report on 10 Recent Cases

**DOI:** 10.3390/v16050709

**Published:** 2024-04-29

**Authors:** Lea Reemann, Nikolaus Kneidinger, Bernd Sczepanski, Andreas Rembert Koczulla

**Affiliations:** 1Institute for Pulmonary Rehabilitation Research, Schoen Klinik Berchtesgadener Land, 83471 Schoenau am Koenigssee, Germany; bsczepanski@schoen-klinik.de (B.S.); rkoczulla@schoen-klinik.de (A.R.K.); 2Department of Medicine V, Comprehensive Pneumology Center Munich (CPC-M), German Center for Lung Research (DZL), Ludwig-Maximilians University (LMU) University Hospital, 81377 Munich, Germany; nikolaus.kneidinger@med.uni-muenchen.de; 3Division of Pulmonology, Department of Internal Medicine, Medical University of Graz, 8010 Graz, Austria; 4Department of Pulmonary Rehabilitation, Universities of Giessen and Marburg Lung Center (UGMLC), German Center for Lung Research (DZL), Philipps-University of Marburg, 35043 Marburg, Germany; 5Teaching Hospital, Paracelsus Medical University Salzburg, 5020 Salzburg, Austria

**Keywords:** COVID-19, lung transplant, immunosuppression, Remdesivir

## Abstract

Due to immunosuppression, transplant recipients are at higher risk of infections with SARS-CoV-2 and worse clinical outcomes than immunocompetent hosts. Furthermore, lung transplant patients represent a special group among solid organ recipients, since pneumonia is the main manifestation of COVID-19. However, data on the course of disease and the changes in morbidity and mortality during the course of the pandemic are limited. In our pulmonary rehabilitation clinic, we treat patients shortly after lung transplant as well as long-term transplant patients. Over the last almost 4 years of pandemic, we witnessed several COVID-19 infections in lung transplant patients in our clinic as well as patients who acquired an infection beforehand. In this paper, we aim at retrospectively describing a series of recent COVID-19 cases in our clinic, looking at the clinical course of disease and outcomes in lung transplant patients.

## 1. Introduction

With the beginning of the COVID-19 pandemic in early 2020 and high morbidity and mortality rates in the general population, the outcomes in immunocompromised patients evoked even more concern [1]. In particular, solid organ transplant recipients (SOTR) are assumed to be at increased risk due to high-level immunosuppression. Among this group of patients, lung transplant (LTX) patients hold a special role, since the allograft is infected directly by the virus.

Immune response to viral infections in immunocompromised patients is distinct from healthy individuals. Immunosuppressive regimens used in SOTRs impair cellular immunity in particular, leading to a reduced viral clearance and increased risk of death from acute infection [2]. Commonly used immunosuppressive drugs to prevent transplant rejection include Calcineurin-inhibitors and Mycophenolate Mofetil (MMF). MMF inhibits the proliferation of both T and B cells by reversibly blocking the enzyme inosinmonophosphatedehydrogenase. In contrast, Calcineurin-inhibitors such as Tacrolimus cause a reduced activation of T lymphocytes only. The combination of B cell-mediated antibody response as well as CD4+ and CD8+ T cells induces effective viral clearance [3]. In patients with hematological malignancies and reduced humoral immune response due to impaired B cell function, CD8+ T cells have been shown to play a crucial role in improving the outcomes of COVID-19 infection [4]. Furthermore, prolonged viral shedding has been associated with a decreased humoral immune response [5,6] as well as intra-host viral evolution [7]. Taking this into account, SOTRs appear to be at a very high risk of a severe course of disease and death, and this has also been reported by many transplant centers at the beginning of the pandemic (cf. discussion).

However, over the course of the pandemic, a lot has changed. Vaccines and antiviral drugs have been introduced, the immune system of the population had the opportunity to adapt to this new pathogen, and new variants of SARS-CoV-2 displaced the wild type. In healthy individuals, vaccination elicits effective neutralizing antibodies [8] as well as a protective T cell response [9]. In SOTRs, humoral as well as cellular response is attenuated. A meta-analysis by Meshram et al. revealed an even lower humoral immune activation in lung transplant patients compared to other SOTRs [10]. As an additional challenge, the rise of new variants such as Omicron increases the risk of breakthrough infections and requires adaptations of existing vaccines [11]. Despite that, vaccination is crucial for reducing morbidity and mortality in this high-risk group. It has been shown that a polyfunctional T cell response after vaccination can at least partly compensate for a poor humoral response [12].

Concerning treatment of SARS-CoV-2 infections in immunocompromised hosts, clinical data are scarce. Tixagevimab/cilgavimab is a combination of antibodies which can be used in pre-exposure prophylaxis, e.g., for patients with a very limited response to vaccination [2]. Remdesivir is a therapeutic option in acute infection, inhibiting the synthesis of viral RNA. In patients with antibody deficiency and chronic or relapsing COVID-19, Remdesivir monotherapy led to a sustained viral clearance in some individuals but was also often associated with treatment failure. In combination with convalescent plasma or monoclonal antibodies, viral clearance was achieved in a considerably higher number of patients [13]. Generally, monoclonal antibodies are another treatment option in the early phase of infection. However, with the development of new viral variants, the neutralizing ability might be impaired [2]. Furthermore, monotherapy with monoclonal antibodies in immunocompromised hosts can possibly induce mutations leading to resistance [14].

Due to the limited number of LTX patients, more reports and, in particular, more recent observations are needed. After 4 years of pandemic and more than 3 years after the first vaccines have been introduced, the immune status in the population and thus also in immunocompromised patients has changed fundamentally. To address this problem, we would like to share some of the experiences we had with COVID-19 infections in lung transplant patients in our clinic for pulmonary rehabilitation. We treat most of the patients who receive a lung transplant in Germany immediately after the acute phase of transplantation in the hospital, as well as long-term transplant patients. In total, more than 3000 LTX patients have been seen in our hospital so far. This case report series of 10 acute SARS-CoV-2 infections in LTX patients presents some more data on the dynamic development of this novel disease in this special group of patients.

## 2. Materials and Methods

Lung transplant patients were hospitalized in our clinic for an inpatient pulmonary rehabilitation program. They received their usual triple-immunosuppressive regimen, which for most patients consists of Tacrolimus, Mycophenolate Mofetil (MMF) and prednisolone. Antigen-tests for COVID-19 were performed in symptomatic patients and in some patients due to routine testing in an outbreak situation. Consulting with the corresponding lung transplant center, patients were treated with Remdesivir if accepted. MMF was usually paused. Development of clinical symptoms was monitored closely along with vital signs, and, if necessary, blood gas analysis (BGA) and laboratory testing was performed. Data presented in this article were obtained during clinical routine care and analyzed retrospectively, and no data were collected solely for scientific purposes. We included LTX patients who developed a COVID-19 infection from September 2023 to January 2024. No other specific inclusion or exclusion criteria were applied. This publication was approved by the Institutional Ethics Committee of the University of Marburg, faculty of medicine, and written informed consent was obtained from all the patients included in this case report series.

## 3. Results

The following table (Table 1) provides an overview of lung transplant patients in our clinic, with the approximate date and type of transplant, underlying disease leading to LTX and age. Details on the course of SARS-CoV-2 infection are provided in the second row, including immune status prior to current infection, clinical symptoms, therapy received and existence of prolonged infection. In general, treatment initiation with Remdesivir is recommended as early as possible after onset of symptoms. However, as the table (Table 1) indicates, treatment was often started with a delay of a couple of days. Reasons included problems with the supply of this drug or delayed reporting of symptoms as well as unspecific symptoms or asymptomatic courses of disease.

To sum up, all of the recent COVID-19 cases in LTX patient we observed in our clinic luckily did not require intensive care or supplemental oxygen. Furthermore, none of the patients experienced a worsening of the baseline lung function. For two of the patients, no lung function prior to infection could be obtained. Some patients showed a higher burden of symptoms, such as chills, fever and fatigue, while others did not show any symptoms at all. Almost all the cases are short-term LTX patients transplanted during the last year. Treatment with Remdesivir was administered in seven out of ten patients. Prolonged infection for more than two weeks was observed in three out of ten patients.

## 4. Discussion

In immunocompetent individuals, morbidity and mortality due to COVID-19 infection are hypothesized to be largely caused by an overshooting inflammatory response with a cytokine release storm. This led to the initial idea that immunosuppressed patients might be less prone to severe courses of disease [2]. However, studies soon showed higher rates of adverse clinical outcomes in patients with, e.g., long-term steroid therapy, hematological malignancies and lymphopenia, suggesting immunosuppression as a disadvantage in SARS-CoV-2 infection [15,16]. A meta-analysis of clinical characteristics in SOTRs found a higher risk for worse outcomes in this group of patients, with the highest mortality rate in the subgroup of lung transplant patients [17]. However, there is some conflicting evidence. Another meta-analysis showed a higher rate of admission to the ICU and acute kidney failure, but no increase in mortality rate, compared to the general population when adjustment was made for demographic and clinical features as well as COVID-19 severity [18].

Various lung transplant centers published data on the clinical outcomes of COVID-19 infection in their patients. Saez-Giménez et al. reported 44 cases in Spain between March and April 2020, of whom 17 patients died and 84% needed respiratory support. This corresponds to higher mortality rates in LTX patients compared to the general population. Furthermore, the prognosis was worse when chronic lung allograft dysfunction (CLAD) preexisted [19]. In the US, one study included 103 LTX patients who tested positive for COVID-19 between May 2020 and March 2022, with a mortality rate of 10% and a hospitalization rate of 24% [20]. Another study only investigated survivors of the infection, showing a significant portion of patients with persistent allograft injury [21]. In 35 patients infected with SARS-CoV-2 in France between March and May 2020, Messika et al. observed a hospitalization rate of 88.6%, 41.9% had to be admitted to the ICU and 14.3% died [22]. In Germany, a study with 31 COVID-19 patients even showed a mortality rate of 39%, and in the survivors a decline in TLC, DLCO and exercise capacity could be seen. Additionally, CLAD was associated with inferior outcomes [23].

Heldman et al. conducted a study on SOTR that generally investigated changing trends in mortality between early vs. late 2020. Here, a decline in the hospitalization rate (75.8% vs. 58.9%) and 28-day mortality (19.6% vs. 13.7%) could be observed [24]. In a nationwide German study, a reduction of mortality could be observed, but the outcome remained unfavorable in the Omicron era [25].

In contrast to the beginning of the pandemic, vaccines are now widely available. In immunosuppressed patients, sufficient dosing is crucial in order to elicit a more effective immune response. Shostak et al. could observe an increase in the overall seropositivity rate from 18% to 42.2% and a rise in antibody titer after administration of a third vaccine dose in LTX patients [26]. Also, a fourth dose has been shown to be beneficial in SOTRs [27]. Furthermore, vaccination is recommended to be administered before transplant. After immunosuppression has been tapered, booster vaccination is recommended as well [28]. Here, the use of corticosteroids [28] as well as MMF [26] could lead to a lower antibody response. Furthermore, regarding different immunosuppressive drugs, Scharringa et al. found that Tacrolimus and Sirolimus could be beneficial in the course of COVID-19. Data on Mycophenolate Mofetil is inconclusive, and it is recommended that one stop its use during acute infection [28].

The recent COVID-19 cases in lung transplant patients that we saw in our clinic did not show any major complications. This corresponds to the development in the general population of a massive decline in mortality and hospitalization rates due to SARS-CoV-2 infection. Recently transplanted patients have usually already had several antigen contacts prior to transplantation via vaccination and/or infection. This seems to be the main reason for the benign courses we observed in our patients in comparison to the more adverse outcomes observed in the beginning of the pandemic.

However, due to the considerable immunosuppression, especially in the early phase after transplantation, antiviral drugs such as Remdesivir are frequently used to prevent severe courses of disease and prolonged viral shedding [29]. Despite treatment, prolonged spreading occurs in several cases and was present in three patients of our cohort. One patient who was already transplanted 25 years ago received four vaccinations and was infected once before his current infection. Thus, this patient is the only patient included in this article who did not undergo immunization prior to transplant. Also, in this case, symptoms were very mild, and he was tested negative 5 days after his initial positive test. The question of which factors determine a prolonged course of COVID-19 in immunosuppressed SOTRs remains unclear. However, the three patients who did not receive Remdesivir did not show a worse clinical course of disease or prolonged viral shedding. This may indicate a limited benefit of the use of antiviral drugs in transplant patients with prior infection or vaccination. However, since this article presents only a series of case reports with a small number of patients, a sound statistical conclusion cannot be drawn. As another treatment option in high-risk patients, the antiviral medication Nirmatrelvir/Ritonavir (brand name Paxlovid) is regarded with caution in SOTRs because of its extensive interactions with immunosuppressive drugs. Pre-exposure prophylaxis (PrEP) and treatment with monoclonal antibodies has proposed fewer infections in a cohort of LTX patients. When infection occurred, the outcome was similar to patients who did not receive PrEP [30]. Treatment with monoclonal antibodies has recently been regarded more critically due to a reduced effectiveness against the current circulating variants and the possible induction of resistance [26]. The difficulties in treating SARS-CoV-2 emphasize even more the importance of an effective prevention via vaccination.

Taken together, the recent observations we made indicate a considerably milder course of disease in LTX patients compared to the beginning of the pandemic. This is probably associated more with prior antigen contacts via infection and/or vaccination and less with the treatment with antiviral drugs. Still, treatment recommendations for LTX patients who tested positive for COVID-19 need further systematic investigations in order to elucidate an optimized course of action. Regular vaccination will continue to play a central role in the prevention of morbidity and mortality in high-risk patients.

## Figures and Tables

**Table 1 viruses-16-00709-t001:** Lung transplant recipients who tested positive for COVID-19 between September 2023 and January 2024.

Prior Immune Status	Clinical Symptoms	Remdesivir?	Prolonged Infection > 14 d?
Combined heart-lung-Tx 1998 due to primary pulmonary hypertension, 34 y. old			
4× vaccination, 1× infection	Mild symptoms of a cold	No	No
Bilateral LTX 9/2023 due to pulmonary fibrosis, 58 y. old			
3× vaccination, 1× infection	No symptoms	Yes, start appr. 5 days after positive test	Yes
Bilateral LTX 8/2023 due to lymphangioleiomyomatosis, 49 y. old			
4× vaccination	Mild fatigue	Yes, start approx. 1 day after positive test, treatment for 10 days	Yes
Bilateral LTX 10/2023 due to interstitial lung disease with systemic sclerosis, 54 y. old			
4× vaccination, 1× infection	Mild symptoms of a cold	Yes, start approx. 5 days after onset of symptoms	No
Bilateral LTX 10/2023 due to chronic obstructive pulmonary disease, 60 y. old			
3× vaccination	No symptoms	No	No
Bilateral LTX 08/2023 due to pulmonary fibrosis, 54 y. old			
2× vaccination, 1× infection	No symptoms	No	No
Bilateral LTX 10/2023 due to interstitial lung disease with amyloidosis, 59 y. old			
3× vaccination, 1× infection	Fatigue, limb pain, sore throat, cough	Yes, start 1 day after onset of symptoms	No
Bilateral LTX 2023 *^1^ due to chronic obstructive pulmonary disease, 63 y. old			
3× vaccination, 1× infection	Cough, fever, vomiting *^2^	Yes, start 4 days after positive test	No
Bilateral LTX 12/2023 due to non-CF bronchiectasis, 35 y. old			
4× vaccination, 2× infection	Fatigue, chills	Yes, start 4 days after positive test	Yes
Right-side single-LTX 11/2023 due to pulmonary fibrosis, 67 y. old			
5× vaccination	No symptoms	Yes, start approx. 4 days after onset of symptoms	No

*^1^ Patient only consented with the year of LTX being published. *^2^ History of nausea and occasional vomiting already prior to infection.

## Data Availability

The datasets presented in this article were obtained during clinical routine care and are not readily available because of privacy protection of our patients.

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
