# Peer review of "COVID-19 in Lung Transplant Recipients: A Report on 10 Recent Cases"

_viruses, 2024, doi:10.3390/v16050709_

Round 1

Reviewer 1 Report

Comments and Suggestions for Authors

The revision of the case series entitled

COVID-19 in lung transplant recipients: a report on 10 recent cases

The article concerns clinical course of COVID-19 diagnosed in the post-pandemic era,  in lung transplant (LTX) recipients.

 In my opinion, the article needs major revision to be published.

1.      The introduction includes the data that could be moved into the discussion. I would suggest to describe in the introduction (in brief) the immunological response to COVID-19 in patients with immunosuppression comparing to healthy population.

2.      The methods should contain either the information of the Ethical Committee agreement to publish the patients’ data or the information concerning the patients’ written informed consent for publication.

3.      The patients’ data lack: the information of lung disease leading to respiratory insufficiency and subsequently to LTX, type of LTX, the data concerning the degree of immunosuppression - eg lymphocytosis, and immunoglobulins levels at the time of COVID-19 infection, the data concerning the results of chest X-ray or chest CT at the time of COVID-19 infection

4.      In discussion I would suggest to refer to other publications concerning COVID-19 course in LTX recipients (the data included in the introduction). The important issue is: whether the benign course of COVID-19 in the presented group was combined with prior COVID-19 infection in some patients, with vaccination against COVID-19, with remdesivir therapy used in some patients?  On this occasion it is worth to discuss the recommendations for vaccination schedule and antiviral therapy in patients with immunosuppression.

5.      I’m not sure if COVID-19 course is worse in patients with immunosuppression. The literature data indicate that cytokine storm is less frequent in such patients, comparing to healthy  population. This is also the point for the discussion.

Author Response

Dear reviewer,

Thank you very much for your suggestions. Here you find my correspondent replies.

  1. The introduction includes the data that could be moved into the discussion. I would suggest to describe in the introduction (in brief) the immunological response to COVID-19 in patients with immunosuppression comparing to healthy population.

Thank you for this input, I added some data on the immune response in immunocompromised patients to the introduction and moved those other parts of the introduction into the discussion.

  1. The methods should contain either the information of the Ethical Committee agreement to publish the patients’ data or the information concerning the patients’ written informed consent for publication.

I included both statements in the methods-part.

  1. The patients’ data lack: the information of lung disease leading to respiratory insufficiency and subsequently to LTX, type of LTX, the data concerning the degree of immunosuppression - eg lymphocytosis, and immunoglobulins levels at the time of COVID-19 infection, the data concerning the results of chest X-ray or chest CT at the time of COVID-19 infection

I now included information on underlying lung disease and type of LTX. Unfortunately, since our article presents only a series of case reports, the clinical data included was collected during routine care. So, we did not assess immunoglobulin levels and did not perform imaging. Our patients were all in a stable respiratory state, so we decided that exposure to radiation was not necessary in those cases. Thus unfortunately, I am not able to add the rest of the information.

  1. In discussion I would suggest to refer to other publications concerning COVID-19 course in LTX recipients (the data included in the introduction). The important issue is: whether the benign course of COVID-19 in the presented group was combined with prior COVID-19 infection in some patients, with vaccination against COVID-19, with remdesivir therapy used in some patients?  On this occasion it is worth to discuss the recommendations for vaccination schedule and antiviral therapy in patients with immunosuppression. 

I moved those publications into the discussion part and I further extended the discussion trying to adequately address those questions.

  1. I’m not sure if COVID-19 course is worse in patients with immunosuppression. The literature data indicate that cytokine storm is less frequent in such patients, comparing to healthy  population. This is also the point for the discussion.

Thank you for this input. I also found some contradictory data and now tried to address this issue in the discussion as well.

Again, I thank you a lot for your highly appreciated feedback and I very much hope that I was able to address all the points to your satisfaction.

With kind regards,

Lea Reemann

Reviewer 2 Report

Comments and Suggestions for Authors

The manuscript entitled “COVID-19 in lung transplant recipients: a report on 10 recent cases” by Lea Reemann et al describes 10 COVID-19 patients that are lung transplant recipients. The immune system of these patients is heavily suppressed, which makes them highly vulnerable to respiratory infections, specifically to SARS-CoV-2. The data presented in the manuscript are of importance to the scientific community, considering the potential for severe disease in these patients.

Altogether I find the manuscript suitable for publication as a case report, please address the following points:

1.       What was the time frame for Remdesivir treatment, i.e. when was the treatment started after the onset of disease symptoms?

2.       Please address the importance of vaccination for lowering disease symptoms, with respect to other published data. Specifically, please discuss T cell immunity since it is essential for limiting disease progression. There are several studies addressing it, such as https://doi.org/10.3390/vaccines11040799

3.       please discuss shortly the importance of antibody treatment, specifically in light of emerging viral variants

4.       Please emphasize what was the criteria for choosing these 10 representative patients, is being a recent case the only criteria?  

Author Response

Dear reviewer,

Thank you very much for your suggestions. Here you find my correspondent replies.

  1. What was the time frame for Remdesivir treatment, i.e. when was the treatment started after the onset of disease symptoms?

I now included that information in the table.

  1. Please address the importance of vaccination for lowering disease symptoms, with respect to other published data. Specifically, please discuss T cell immunity since it is essential for limiting disease progression. There are several studies addressing it, such as https://doi.org/10.3390/vaccines11040799

Thank you for this input, I added some more data on vaccination and T cell immunity in the introduction as well as in the discussion. I hope now I could adequately address this topic.

  1. please discuss shortly the importance of antibody treatment, specifically in light of emerging viral variants

Concerning this topic I now included some data in the introduction as well as in the discussion.

  1. Please emphasize what was the criteria for choosing these 10 representative patients, is being a recent case the only criteria?

Yes it basically is the only criteria, I included a short comment in the method section.

Again, I thank you a lot for your highly appreciated feedback and I very much hope that I was able to address all the points to your satisfaction.

With kind regards,

Lea Reemann

Round 2

Reviewer 1 Report

Comments and Suggestions for Authors

Thank you very much for the extensive revision of the text, according to my remarks.  I accept the current version of the manuscript for publication.